# RGB-EVENT MOT: A CROSS-MODAL BENCHMARK FOR MULTI-OBJECT TRACKING

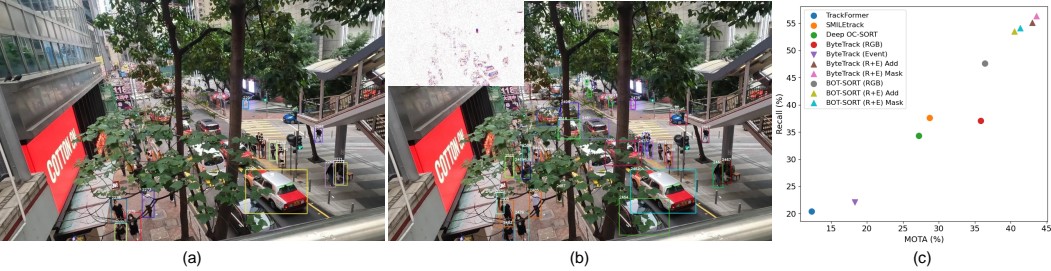

Figure 1: Illustration of sequence samples from our RGB-Event MOT benchmark. (a) RGB-based MOT results. (b) Our RGB-Event MOT results. (c) Quantitative comparisons of RGB *vs.* RGB-Event methods.

## ABSTRACT

Leveraging the power of contemporary deep learning techniques, it has become increasingly convenient for methodologies to recognize, detect, and track objects in real-world scenarios. Nonetheless, challenges persist, particularly regarding the robustness of these models in recognizing small objects, operating in low-illumination conditions, or dealing with occlusions. Recognizing the unique advantages offered by Event-based vision - including superior temporal resolution, vast dynamic range, and minimal latency - it is quickly becoming a coveted tool among computer vision researchers. To bolster foundational research in areas such as object detection and tracking, we present the first cross-modal RGB-Event multi-object tracking benchmark dataset. This expansive repository encompasses nearly one million carefully annotated ground-truth bounding boxes, offering an extensive data resource for research endeavors. Designed to augment the practical implementation of Event-based vision technology, this dataset proves particularly beneficial in intricate and challenging environments, including low-light situations, scenarios marked by occlusions, and contexts involving diminutive objects. The utility and potency of cross-modal detection and tracking models have been extensively tested and confirmed through our experimental studies. The encouraging results not only affirm the necessity of these models but also highlight their efficacy, thus emphasizing the benchmark's potential to significantly propel the advancement of Event-based vision technology. We have included the code in the supplementary material and will make the dataset publicly available.

## 1 INTRODUCTION

In the realm of computer vision, RGB image-based learning algorithms have made significant strides in multi-object tracking (MOT) under normal environments. However, challenges persist in detecting and tracking objects under less-than-ideal conditions, such as low-light environments, occluded scenes, and situations involving small or distant objects. For example, as illustrated in Table 1, a substantial performance decline is observed when transitioning from MOT17 (Milan et al., 2016) to our dataset, which is inclusive of varied corner cases.

Table 1: Comparisons of tracking performance of state-of-the-art RGB-based MOT methods on the well-known MOT benchmark dataset, i.e., MOT17, and our dataset with varied corner cases. For the two metrics, the larger, the better. Note that even though we trained the methods with the training set of our dataset, the performance is still much lower than that on MOT17 (see the results in Table 4).

| Dataset | Metric | TrackFormer | SMILEtrack | BoT-SORT | ByteTrack | Deep OC-SORT |
|---------|--------|-------------|------------|----------|-----------|--------------|
| MOT17   | MOTA   | 74.1        | 81.0       | 80.5     | 80.3      | 79.4         |
|         | IDF1   | 68.0        | 80.5       | 80.2     | 77.3      | 80.6         |
| Ours    | MOTA   | 12.3        | 28.7       | 28.8     | 28.2      | 27.2         |
|         | IDF1   | 15.9        | 34.5       | 34.4     | 33.6      | 33.7         |

**Event cameras**, with their high dynamic range and low latency, offer significant advantages in these challenging scenarios. They are capable of naturally capturing object patterns in low-light situations, overcoming one of the major limitations of traditional RGB cameras. Furthermore, the distinctive sensing pattern of Event cameras makes them an invaluable tool for sensing variations in occluded objects, another hurdle in RGB-only sensing patterns. Moreover, for the detection and tracking of small or remote objects, Event cameras can provide extra motion clues to help build effective solutions. Their unique capabilities allow for the observation and interpretation of subtle changes (Perot et al., 2020), making them particularly suited for tasks that require precision and sensitivity. *Therefore, it is promising to address the above-mentioned corner cases by developing RGB-Event-based MOT,* as illustrated in Fig. 1.

However, there are limited datasets available for current algorithm design and validation. Thus, we make the *first attempt* to establish a benchmark that leverages the synergies of RGB and Event-based data for MOT. This paper provides an in-depth exploration of our methodology, empirical evidence of its superiority over traditional RGB-only sensing, and a roadmap for further research and development in this domain. Moreover, we have built a baseline to validate the superiority of cross-modal RGBEvent perception. Our findings lay the groundwork for a new frontier in cross-modal RGB-Event-based detection and tracking, promising significant improvements in a wide range of applications.

In summary, the main contributions of this paper are two-fold:

- we introduce a cross-modal RGB-Event dataset for MOT. This dataset represents a groundbreaking effort to confront the challenging scenarios frequently encountered in object perception, such as those involving diminutive objects, adverse illumination conditions, and occlusions. The main objective behind this effort is to pave the way for the development of a more robust object perception system; and

- we undertake an exhaustive assessment, encompassing state-of-the-art MOT algorithms. This evaluation was designed to scrutinize the potential advantages of fusing both RGB and event-based data for MOT. Our analysis not only sheds light on the performance enhancements but also provides insights into the potential synergies between the two data modalities, underscoring the practical significance of their integration in advancing the field of MOT.

## 2 RELATED WORK

**RGB image-based perception** has formed the cornerstone of computer vision research for a long time. Classical methods employ handcrafted feature extraction techniques like Scale-Invariant Feature Transform (SIFT) and Histogram of Oriented Gradients (HOG). These have seen considerable success in numerous applications. However, the subsequent rise and rapid progression of deep learning marked a pivotal shift in this field, with Convolutional Neural Networks (CNNs) outclassing previous methods in tasks like object detection, segmentation, and recognition. In recent developments, numerous learning-based MOT methods have illustrated exceptional capabilities in object localization. Deep SORT (Wojke et al., 2017) integrates deep learning-driven appearance descriptors with the classical SORT tracking methodology. Tracktor++ (Bergmann et al., 2019) innovatively employs the object detector for tracking, removing the necessity for data association. FairMOT (Zhang et al.,

2021), a one-shot approach, concurrently detects and tracks objects, addressing challenges inherent in two-shot methods. CenterTrack (Zhou et al., 2020) extends object detection to video object tracking by using the object as the center point. TrackR-CNN (Voigtlaender et al., 2019) expands upon Mask R-CNN (He et al., 2017) to accommodate video object detection and tracking tasks, and ByteTrack (Zhang et al., 2022b) executes MOT through a detection and re-identification pipeline.

Frameworks like Faster R-CNN (Girshick, 2015), YOLO (Redmon et al., 2016), and SSD (Liu et al., 2016) have revolutionized the field of object detection by adopting region proposal networks and end-to-end detection paradigms. Additionally, semantic segmentation witnessed significant enhancements by introducing techniques like Fully Convolutional Networks (FCNs) and U-Nets. Despite these advancements, traditional image-based perception has limitations, particularly in the context of real-time processing, low-illumination scenarios, and dealing with occluded or minuscule objects. While several algorithms (Dong et al., 2015; Hou et al., 2023; Li et al., 2023) have been introduced to enhance image quality, subsequently facilitating high-level visual tasks, these methods often inevitably extend computation times for image reconstruction. Furthermore, there is no guarantee that the reconstructed images will faithfully represent the authentic visual characteristics of the scenes. Hence, it becomes imperative to integrate supplementary visual cues or signals to interpret the scene accurately.

**Event-based perception**, on the other hand, is a relatively newer development in the field of computer vision, particularly driven by the emergence of event-based sensors. These sensors, such as Dynamic Vision Sensors (DVS), offer substantial advantages in terms of high temporal resolution, wide dynamic range, and low latency. They operate by capturing changes in pixel intensity, producing a stream of Events that are timestamped with high precision.

This unique operating principle allows event-based perception to excel when traditional image-based perception falls short - for instance, in low-light conditions, high-speed scenarios, or environments with high dynamic range. Nevertheless, the major challenge for Event-based perception has been the scarcity of rich, well-annotated datasets needed for training robust models.

Recent efforts have been made to curate more extensive and varied datasets for Event-based perception, and researchers have also started exploring ways to apply popular deep learning techniques, like CNNs, to Event data. In an exploration of Event-based sensing, Bryner et al. (2019) endeavored to ascertain the 6-DOF pose of a camera, making significant strides in understanding the application of Event cameras. In a complementary vein, Mitrokhin et al. (2018) proposed a unique solution to accommodate camera motion, employing a parametric model that captures the intricate spatio-temporal geometry of Event data. Li & Shi (2019), in an effort to enrich the understanding of Event-stream object appearance, incorporated the VGG-Net-16 into their methodology, thereby demonstrating a robust approach to Event-based object tracking. de Tournemire et al. (2020) proposed the first large-scale object detection dataset for automotive applications. Moreover, Perot et al. (2020) further proposed an Event-based object detection dataset with a high spatial resolution Prophesee Event camera. Drawing inspiration from the well-established Siamese-matching paradigm, Chae et al. (2021) developed an innovative solution for object tracking that learns an edge-aware similarity within the event domain. Building on the foundational tracking-learning-detection pipeline, Ramesh et al. (2018) ventured into the development of an object tracking algorithm specifically designed for Event cameras. This work notably represented the first foray into the realm of learning-based long-term Event tracking. In recent developments, Zhang et al. (2022a) introduced a pioneering spiking Transformer, designed to encode the spatio-temporal information of object tracking. Extending the potential of Event data, Zhu et al. (2022) proposed the utilization of inherent motion information within Event data as a strategy to achieve effective object tracking, marking a substantial advancement in the field. To facilitate the fusion of RGB and event data, Zhu et al. (2023) proposed to utilize augmentations on transformer attention matrix patterns, resulting in an effective cross-modal fusion manner. Zubić et al. (2023) explored the effectiveness of different representations on event-based object detection.

However, there's still much ground to be covered in developing more sophisticated algorithms and models that can effectively utilize the high-temporal resolution data generated by event-based sensors. Our proposed cross-modal RGB-Event benchmark is an attempt to address these challenges and advance the field of event-based perception.

Table 2: Statistic of multi-object tracking and cross-modal RGB-Event tracking datasets, where "RGB" denotes the RGB images, "Event" indicates the Event streams and "GS" represents the grayscale images.

| Dataset | MOT17 | MOT20 | DanceTrack | FE108 | VisEvent | COESOT | RGBEvt-MOT |
|---|---|---|---|---|---|---|---|
| Videos | 14 | 8 | 100 | 108 | 820 | 1,354 | 12 |
| Avg. tracks | 96 | 432 | 9 | 1 | 1 | 1 | 247 |
| Total. tracks | 1,342 | 3,456 | 990 | 108 | 820 | 1,354 | 2,962 |
| Avg. len. (s) | 35.4 | 66.8 | 52.9 | 48.3 | 18.1 | 14.14 | 67.8 |
| Total len. (s) | 463 | 535 | 5292 | 5216 | 14,845 | 19,148 | 813 |
| Modalities | RGB | RGB | RGB | GS+Event | RGB+Event | RGB+Event | RGB+Event |
| Resolution | 1920×1080 | 1920×1080 | – | 346×260 | 346×260 | 346×260 | 2560×1600 |
| Sensor Latency (ms) | 33 | 40 | 50 | $1e^{-3}$ | $1e^{-3}$ | $1e^{-3}$ | $1e^{-3}$ |
| Total images | 11,235 | 13,410 | 105,855 | 208,672 | 371,127 | 478,721 | 21,336 |

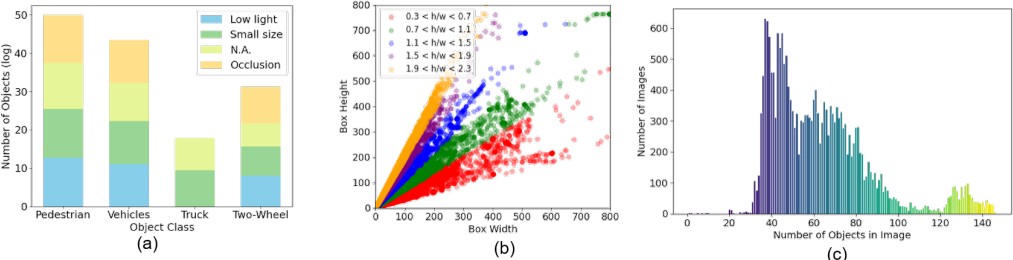

Figure 2: Statistics of our proposed RGB-Event MOT benchmark. (a) The number of objects with different classes and attributes. (b) Size distribution of object bounding boxes. (c) Distribution of the number of bounding boxes per image.

## 3 DETAILS OF OUR RGB-EVENT BENCHMARK DATASET

As aforementioned, current learning-based methods, equipped with exceptional modeling capacities, have demonstrated proficient performance in visual object recognition under normal conditions. However, their effectiveness is severely compromised under complex, real-world scenarios. This observation prompts an exploration into supplementary vision cues to augment tracker robustness. We are particularly drawn to the untapped potential of event cameras, characterized by rich temporal/motion data and a high-dynamic range. Leveraging this, we aim to enhance object recognition under adverse visual scenarios marked by small sizes, low illumination, and occlusions. Our dataset hinges on the distinctive advantages of event data, enabling algorithms to gain comprehensive insights into the surrounding environment. The detailed comparisons between related datasets are shown in Table 2.

### 3.1 DATASET COLLECTION

In the process of dataset collection, we strategically employed a stereo configuration of RGB and event cameras to capture image and event streams. This methodology served to create a dataset exclusively comprised of fully static viewpoints, thereby guaranteeing consistency across all data points. The focus of our data collection was to encapsulate challenging scenarios that are typically experienced in the field, such as low-light environments, small-sized objects, and instances of occlusion.

The core of our data collection procedure was underscored by the crucial alignment task. In accordance with calibration methods delineated by (Zhang, 2000), we initially attempted alignment by matching planar patterns across views to compute the homograph matrix. However, the inherent limitations of stereo parallax presented a challenge; it restricted the accuracy of alignment to a specific depth range.

In response to this obstacle, we embraced a more hands-on approach to ensure the precision of alignment across stereo views. This involved the manual collection of correspondence points, from

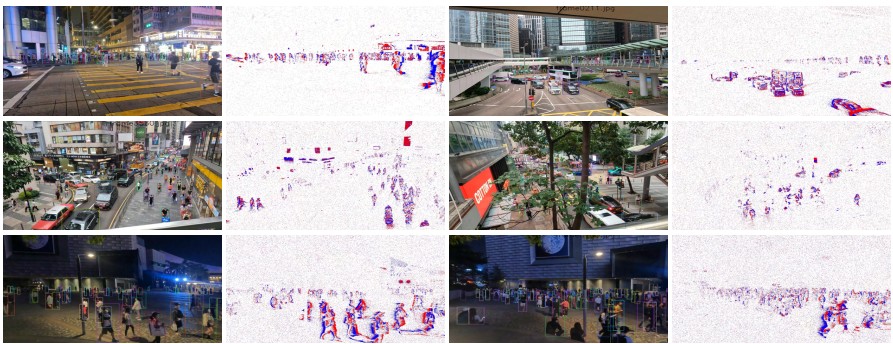

Figure 3: Sequence samples from our RGB-Event MOT benchmark.

which we calculated the homograph matrix. Our meticulous approach to this critical step enhanced the reliability of the alignment, fostering the robustness and utility of our dataset. This refinement to our methodology ensured desirable alignment for stereo views, thereby fortifying the dependability of our dataset and its relevance to our study.

It should be highlighted that due to the intrinsic parallax between the stereo of the event and RGB data, achieving an exact correspondence between them is challenging. In contrast to cross-modal reconstruction tasks, which necessitate stringent alignment across diverse data modalities, high-level computational tasks, such as detection and tracking, often remain robust even with minor spatial discrepancies in certain regions or boundaries. Consequently, the methodology proposed for data collection is instrumental in facilitating studies on visual perception tasks.

## 3.2 RGB-EVENT SEQUENCES

The comprehensive comparison and statistics of our proposed dataset are presented in Tables 3 and A-1. Comprising 12 unique sequences, each featuring RGB frames paired with a corresponding Event stream, the dataset represents a substantive resource for advancing the study of MOT. The RGB data is captured using the GoPro-10 camera, boasting a formidable 4K resolution of $3840 \times 2160$. Meanwhile, the Event stream is recorded utilizing a Celex5 event camera, offering a spatial resolution of $1280 \times 800$. The meticulous calibration of both cameras yields an RGB stream operating at 24-30 frames per second with a resolution of $3840 \times 2160$, synchronized with an Event stream at a resolution of $1280 \times 800$. Our dataset specifically addresses three prevailing challenges that traditional RGB cameras struggle to manage effectively: objects of small size, low-illumination conditions, and occlusion.

**Small Size Objects.** The task of discerning small objects poses a considerable challenge in the domain of object detection, a hurdle that traditional RGB cameras often struggle to overcome due to the paucity of semantic information inherent in smaller objects. This issue may be addressed effectively by exploiting the temporal information of the objects, which can offer valuable insights for object recognition. With this in mind, our dataset incorporates scenarios featuring small-sized objects, thereby fostering the development of sophisticated algorithms capable of leveraging the rich temporal information present in event data for more accurate detection and tracking of diminutive entities.

Table 3: Details of our RGB-Event benchmark dataset for MOT.

| #Seq. | FPS (RGB) | Length | Track | Density | Attributes | Training |
|-------|-----------|--------|-------|---------|------------|----------|
| 01 | 24 | 2249 | 419 | 77.6 | Small-size | ✓ |
| 02 | 24 | 965 | 174 | 53.7 | Low-light | ✓ |
| 03 | 24 | 1450 | 292 | 56.8 | Low-light | ✗ |
| 04 | 24 | 1675 | 316 | 63.1 | N.A. | ✓ |
| 05 | 24 | 1401 | 273 | 130.1 | Small-size | ✗ |
| 06 | 24 | 1411 | 181 | 73.1 | Low-light | ✗ |
| 07 | 24 | 1276 | 210 | 70.0 | Occlusion | ✗ |
| 08 | 24 | 1793 | 425 | 84.7 | Low-light | ✓ |
| 09 | 30 | 799 | 211 | 65.2 | N.A. | ✓ |
| 10 | 30 | 1471 | 242 | 47.0 | N.A. | ✗ |
| 11 | 30 | 2999 | 108 | 38.3 | Occlusion | ✓ |
| 12 | 30 | 3000 | 111 | 44.2 | Occlusion | ✗ |

**Low-Illumination Condition.** In practical applications, diminished lighting conditions often lead to a compromised image quality, which in turn negatively impacts the accuracy of sensory data. Owing to the high dynamic range of event cameras, these devices serve as an effective solution to such challenging circumstances. Our dataset deliberately includes scenarios captured under low-illumination conditions. This inclusion presents an opportunity for algorithms to augment their performance in less-than-ideal lighting environments, thus serving as a robust test bed for assessing their adaptability to real-world conditions.

**Occlusion.** Target objects often encounter partial or complete obstruction by other entities. This interference is particularly challenging for tracking systems and can impact the functionality of neural networks such as Multiple Layer Perceptrons and CNNs. Specifically, it may cause the object's features to be contaminated by the obstructive information, leading to incorrect or missed detections. However, thanks to the distinctive sensing pattern of the event camera, changes can be detected with extremely low latency. This capability helps the network concentrate on the subject matter instead of the obstructions, thereby reducing the likelihood of feature contamination. To corroborate this premise, we have included sequences in our dataset that present varying degrees of occlusion. This encourages the creation of tracking and detection mechanisms that are adept at managing such circumstances effectively.

## 3.3 Data Format

**Event Stream**. Due to the fact that the event data is sparse in the 3D spatio-temporal space, we provide the raw data of event clouds. It contains four dimensions $(x, y, p, t)$, where $x \in [1, 1280], y \in [1, 800]$ denote the spatial locations, $p \in \{-1, 1\}$ indicates the polarity, i.e., increasing or decreasing of the pixel intensity and $t$ is the timestamp with the unit of $\mu s$.

Besides, we also provide frame-based representation to quickly adapt the event data into an image processing pipeline. Specifically, we equally quantify the time between two frames into three intervals $\delta_t$. For each pixel of an event frame, its value is calculated by the accumulation of

$$I(x, y) = \bar{I} + \delta \times \sum_{t_i} e(x, y, t_i), t_i \in \delta_t, \tag{1}$$

where $I(x, y) \in \mathbb{R}^{h,w}$ indicate the event frame. By employing voxelization in the temporal domain, and $\bar{I}$, $\delta$ are scalars representing the base and increment magnitude, which are empirically selected as 127 and 30, respectively. We can effectively standardize the sporadically distributed event data, making it more amenable for neural network processing. This approach allows us to leverage established CNN architectures for event data processing. It is important to acknowledge, however, that this methodology might intrinsically reduce the high temporal resolution inherent to event data. Nonetheless, the primary objective of this paper is to establish a benchmark for evaluating RGB-Event MOT algorithms, and our experimental validation serves as an assessment of the effectiveness of integrating event data into RGB-centric vision systems.

**RGB Stream**. Similar to the image-based MOT, the RGB stream is structured as a sequence of images. Due to the unique structure of event data, we have aligned the images to the event stream via homograph transformation.

**Annotation**. We adhere to the MOT20 annotation format, which includes ten elements: frame index, object ID, bounding box coordinates $(x, y)$ with its width $(w)$ and height $(h)$, confidence score, two placeholder values (-1, -1), and the object class. Our dataset contains annotations for four different categories of objects, namely pedestrians, vehicles, trucks, and two-wheels.

## 4 Baseline Methods

In order to leverage the unique characteristics of event data, i.e., the high-dynamic range and temporal resolution, for facilitating image-based object recognition, we have developed a cross-modal pipeline to conduct an in-depth analysis of cross-modal fusion. We initiate our discussion with a comprehensive exploration and evaluation of the prevailing RGB-based MOT algorithms. The existing MOT algorithms and datasets predominantly concentrate on distinct categories, for instance, pedestrians or vehicles. This specialization typically leads to a bifurcated tracking pipeline that ini-

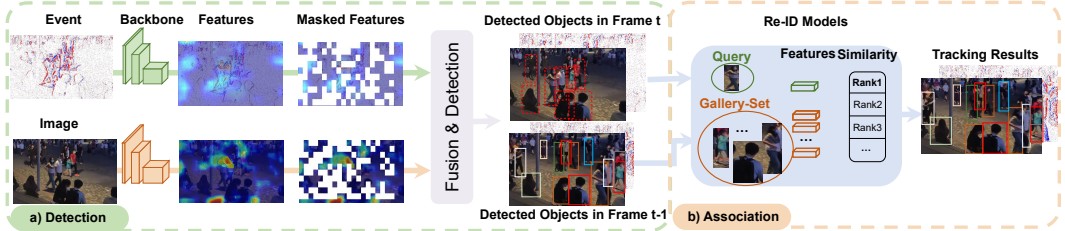

Figure 4: The flowchart of our proposed RGB-Event MOT framework. The process commences with the incorporation of both RGB and event streams as input, which are then processed through a typical backbone neural network to facilitate primary feature embedding. A fusion strategy (such as simply averaging of dual streams or feature mask modeling) is then administered to the cross-modal data, amplifying their interactive capabilities, and the fused features are further fed into a typical RGB image-based detector. In the **association** stage, we can employ a re-identification module to associate boxes with distinct targets. **Note** that the mask modeling is only applied in the training phase.

tially identifies the objects and subsequently associates distinct bounding boxes with corresponding IDs from preceding frames.

Consequently, two avenues for enhancing MOT accuracy emerge: (**1**) optimizing detection performance through the integration of RGB and event data and (**2**) refining the association mechanism. Given that the quality of detection proposals significantly influences the overall MOT performance, our paper predominantly concentrates on augmenting the detection outcomes by amalgamating RGB and event branches, aiming to elevate the overall efficacy of MOT.

Given the extensive spatial dimensions inherent to both image and event data, it is imperative to initiate the fusion of these two kinds of data streams at an early stage to ensure algorithmic efficiency. As depicted in Fig. 4, our initial step involves the extraction of primary feature maps from both RGB and Event streams. Subsequently, to manage a highly effective fusion of RGB and Event streams, we investigate different fusion techniques: (1)"**AVG**": directly averaging the event and RGB streams to harness the collective information; (2)"**MASK**": introducing an advanced mask modeling to facilitate the network's adaptive utilization of both RGB and Event data.

These fused features are then directed through the extraction and proposal network to generate object proposals. In the concluding step, Re-ID or alternative algorithms are employed to associate these proposals, culminating in the achievement of MOT.

## 5 EXPERIMENTS

In light of the preceding discussion, we have conducted a series of experiments to corroborate the necessity of integrating event data in such temporal contexts. Our initial evaluation involved assessing the performance of **pre-trained** MOT trackers, including TrackFormer (Meinhardt et al., 2022), SMILEtrack (Wang et al., 2023), BoT-SORT (Aharon et al., 2022), ByteTrack (Zhang et al., 2022b), and Deep OC-SORT (Maggiolino et al., 2023), on our specifically curated datasets, to gauge their efficacy under extreme conditions.

### 5.1 EVALUATION METRICS

Evaluating the performance of MOT algorithms requires several specialized metrics that capture various aspects of tracking performance. Following the previous works (Milan et al., 2016; Dendorfer et al., 2020), we utilize the following metrics for the MOT task.

**Multiple Object Tracking Accuracy (MOTA)** is a composite measure that takes into account false positives, missed targets, and identity switches. It is defined as $1 - (errors/GT)$, where errors are the sum of false positives, misses, and identity switches, and the GT is the total number of ground truth bounding boxes. MOTA can take values from $-\infty$ to 1, with 1 indicating perfect tracking performance.

Table 4: Quantitative comparisons of different methods: ByteTrack (Zhang et al., 2022b), Deep OC-SORT (Maggiolino et al., 2023) and TrackFormer on the proposed dataset for MOT. Gray region denotes the methods w/o re-training on our dataset. "↑" (resp. "↓") indicates the higher (resp. lower), the better. We did not obtain results when training BOT-SORT only with event data because the training process keeps crashing despite our attempts with different solutions. Such an observation also indicates that straightforwardly adapting existing RGB-based detectors to event data is not an optimal choice, and event data tailored detectors should be investigated.

| Methods | Modality | Fusion | MOTA ↑ | IDF1 ↑ | FP ↓ | FN ↓ | IDs ↓ | Recall ↑ | MT ↑ | ML ↓ |
|---|---|---|---|---|---|---|---|---|---|---|
| TrackFormer | RGB | – | 12.3% | 15.9% | 47,993 | 512,700 | 4,600 | 20.4% | 51 | 983 |
| SMILEtrack | RGB | – | 28.7% | 34.5% | 53,968 | 402,353 | 2,810 | 37.6% | 163 | 634 |
| BoT-SORT | RGB | – | 28.8% | 34.4% | 54,012 | 402,263 | 2,809 | 37.6% | 162 | 634 |
| ByteTrack | RGB | – | 28.2% | 33.6% | 54,948 | 404,724 | 2,801 | 37.2% | 150 | 638 |
| Deep OC-SORT | RGB | – | 27.2% | 33.7% | 43,515 | 423,265 | 2,378 | 34.3% | 136 | 716 |
| ByteTrack | RGB | – | 35.8% | 37.1% | 67,685 | 342,070 | 4,072 | 37.1% | 264 | 543 |
| | Event | – | 18.3% | 22.6% | 22,640 | 502,211 | 1,755 | 22.1% | 78 | 1,225 |
| | RGB+Event | AVG | 43.0% | 41.8% | 73,153 | 289,560 | 4,489 | 55.1% | 327 | 422 |
| | RGB+Event | MASK | 43.6% | 41.3% | 77,413 | 281,297 | 4,825 | 56.3% | 332 | 381 |
| BOT-SORT | RGB | – | 36.4% | 37.9% | 67,933 | 337,852 | 4,131 | 47.6% | 280 | 783 |
| | Event | – | – | – | – | – | – | – | – | – |
| | RGB+Event | AVG | 40.5% | 41.0% | 76,950 | 295,155 | 5,060 | 53.5% | 298 | 460 |
| | RGB+Event | MASK | 41.3% | 41.1% | 77,105 | 295,831 | 5,167 | 54.1% | 301 | 416 |

**IDF1 Score** measures the ratio of correctly identified detections over the average number of ground-truth and computed detections. It provides a balance between precision and recall and is particularly useful in assessing the performance of trackers in handling identity switches.

**Mostly Tracked (MT) and Mostly Lost (ML) Targets** provide measures of the ratio of ground-truth trajectories that are covered by the tracker for at least 80% of their respective lifespans (MT), or less than 20% (ML). High MT and low ML values are desirable.

**Identity Switches (IDS)** counts the number of times the identity of a tracked object is incorrectly changed. Lower values are better, as fewer identity switches indicate more accurate tracking.

**False Positives (FP) and False Negatives (FN)**. FP represents the instances where the tracker mistakenly identifies an object that does not exist in the ground truth. Conversely, FN is the instances where the tracker fails to identify an object that is present in the ground truth. Lower FP and FN values indicate better tracking performance as they signify fewer misidentifications and omissions respectively.

## 5.2 EXPERIMENTAL RESULTS

In addition, we engaged state-of-the-art MOT algorithms, namely ByteTrack (Zhang et al., 2022b) and BOT-SORT (Aharon et al., 2022), applying them to diverse combinations of modalities to ascertain their adaptability and performance metrics. The results, delineated in Table 4, unequivocally underscore three pivotal observations. We also refer readers to Figs. A-2 to A-7 in *Appendix* for more visual results, as well as the *video demo* contained in the *Supplementary Material*.

**Challenges Presented by the Proposed Dataset.** Contemporary MOT trackers typically exhibit exemplary performance, as evidenced by nearly 80% MOTA on established benchmarks like the MOT16/20 datasets. However, when confronted with our proposed dataset, even state-of-the-art methods are significantly challenged, achieving a MOTA of approximately 28%. This precipitous decline in performance underscores both the complexity and the critical need for intensified research efforts to address these nuanced corner cases within the field of computer vision.

**Advancements Afforded by Event Data Integration.** Following the integration of event data into MOT algorithms, a noticeable enhancement in performance is observed, with ByteTrack and BOT-SORT. This uptick underscores the effectiveness of incorporating event data into MOT processing. It's essential to highlight that our exploration into the utilization of RGB-Event MOT is foundational; we anticipate subsequent, more intricate research to further refine and optimize RGB-Event algorithms.

**The Influence of Aggregation Techniques on Data Fusion.** Our experiments involved utilizing simplistic averaging and mask modeling for feature aggregation, revealing nuanced impacts on the outcome. Evidently, mask modeling outperformed the fundamental average fusion, indicating its

efficacy in enhancing performance. These insights are instrumental, prompting the development of sophisticated fusion methods tailored to address the intricacies of cross-modal challenges.

## 6 DISCUSSIONS

Despite the strengths and advantages of our dataset, it is also worth acknowledging the following limitations, which would facilitate the following research:

**Static Viewpoints**. The current version of our dataset solely comprises static/fixed viewpoints. The lack of sequences captured from moving viewpoints may constitute a limitation as it restricts the range of scenarios that our dataset can simulate. Note that in fields such as surveillance, even a fixed viewpoint can be very helpful.

**Isolated Hard Cases**. The dataset includes a variety of hard cases, such as small-sized objects, low-illumination conditions, and occluded objects. However, these hard cases are presented in isolation from each other, providing limited opportunities to evaluate the effectiveness of algorithms in situations where multiple hard cases co-occur. Addressing multiple challenges concurrently is also required in some real-world scenarios.

**Diversity of Scenes**. While our dataset captures a broad range of scenarios, the diversity of scenes, in terms of background settings, lighting conditions, and object types, may be insufficient to challenge and evaluate tracking algorithms fully. Additional diversity would likely further improve the generalizability of the models trained on this dataset. In the future, we will continue to collect data with attributes like motion blur and over-exposure to exploit the potential of event data further.

**Potential Research Directions.** An effective fusion strategy is paramount, as evidenced by experimental results indicating that the application of a mask enhances the representation of cross-modal embeddings. Consequently, there is scope for further exploration and refinement of fusion techniques and regularization terms to optimize this process. Additionally, devising a proficient embedding method for event data is essential, potentially entailing the direct processing of raw event data to preserve its intrinsic high temporal resolution. Furthermore, the development of a specialized box association algorithm is requisite, one that is tailored to capitalize on the unique attributes of event data, thereby bolstering the efficacy of the RGB-Event MOT algorithm. This nuanced approach promises to leverage the distinct characteristics of event data, offering enhanced performance and accuracy in complex tracking environments.

## 7 CONCLUSION

In this paper, we introduced a novel cross-modal RGB-Event dataset for MOT, designed to push the boundaries of current tracking methods. The dataset presents a collection of sequences incorporating challenging scenarios like low illumination, small object detection, and object occlusion, which are difficult for traditional RGB-only sensing methods to handle effectively.

The unique combination of RGB and event streams, captured using state-of-the-art equipment, offers a rich data source for developing and evaluating advanced MOT algorithms. This dataset opens the door to the exploration of the high-dynamic-range and low-latency capabilities of event cameras in tandem with conventional RGB data. While we acknowledge some limitations, including a lack of moving viewpoints and isolated presentation of hard cases, the value of this dataset in advancing the field of MOT is undoubted. We expect it to motivate researchers to develop more robust and versatile detection and tracking methods capable of overcoming the challenges presented.

Moving forward, we plan to address the identified limitations in future versions of our dataset, thereby providing even more comprehensive tools for the development of RGB-Event-based MOT algorithms. By establishing this benchmark, we aim to inspire further research and innovation in the field and anticipate significant advances in the performance and capabilities of MOT systems.

## ETHICS STATEMENT

The development of cross-modal RGB-Event MOT systems offers promising advancements in object detection and tracking, particularly in challenging environments such as low-illumination con-

ditions or those marked by occlusions. By leveraging Event-based vision's distinctive advantages, including superior temporal resolution, vast dynamic range, and minimal latency, our benchmark dataset aims to elevate the standards and robustness of computer vision models. It is our belief that implementing this dataset can lead to safer and more efficient applications in areas that rely on real-time detection and tracking, such as autonomous vehicles and surveillance. This research was carried out using synthesized and publicly available data, thus eliminating the need for live participants and ensuring there are no potential privacy breaches. All original creators of utilized datasets have been duly acknowledged in accordance with academic norms. Furthermore, we are committed to making the code and dataset publicly accessible, furthering transparency and promoting collaborative research in this domain.

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

## A  APPENDIX

In this appendix, we provide more statistics and visual results omitted from the manuscript due to space limitations.

### A.1  DETAILS OF OUR RGB-EVENT DATASET

As shown in Table A-1 and Fig. A-1, we give the samples and detailed statistics of our proposed RGB-Event dataset in the section.

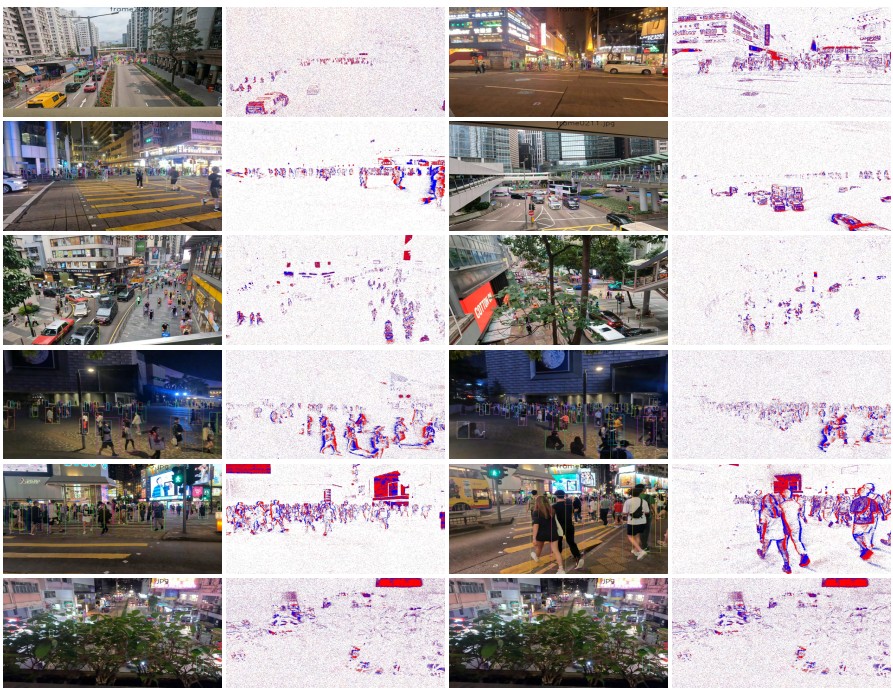

Figure A-1: Illustration of sequence samples from our RGB-Event MOT benchmark.

Table A-1: Statistics of annotations in the proposed cross-modal RGB-Event tracking datasets.

| #Seq. | Sum of a Seq. | | | | | Average of a Seq. | | | | |
|---|---|---|---|---|---|---|---|---|---|---|
| | Pedes. | Vehicle | Truck | Two-Wheels | All cls. | Pedes. | Vehicle | Truck | Two-Wheels | All cls. |
| 01 | 122,670 | 41,366 | 7,149 | 1,792 | 172,977 | 55.03 | 18.56 | 3.21 | 0.80 | 77.60 |
| 02 | 54,779 | 9,688 | 0 | 340 | 64,807 | 45.42 | 8.03 | 0.00 | 0.28 | 53.73 |
| 03 | 65,580 | 15,481 | 0 | 1,296 | 82,357 | 45.23 | 10.68 | 0.00 | 0.89 | 56.80 |
| 04 | 59,973 | 20,145 | 4,032 | 474 | 84,624 | 44.72 | 15.02 | 3.01 | 0.35 | 63.11 |
| 05 | 187,234 | 34,329 | 5,915 | 250 | 227,728 | 106.99 | 19.62 | 3.38 | 0.14 | 130.13 |
| 06 | 124,731 | 36,793 | 2,256 | 1,248 | 165,028 | 55.26 | 16.30 | 1.00 | 0.55 | 73.12 |
| 07 | 77,783 | 0 | 0 | 0 | 77,783 | 60.96 | 0.00 | 0.00 | 0.00 | 60.96 |
| 08 | 94,935 | 0 | 0 | 0 | 94,935 | 84.69 | 0.00 | 0.00 | 0.00 | 84.69 |
| 09 | 52,077 | 0 | 0 | 0 | 52,077 | 65.18 | 0.00 | 0.00 | 0.00 | 65.18 |
| 10 | 66,816 | 2,394 | 0 | 0 | 69,210 | 45.42 | 1.63 | 0.00 | 0.00 | 47.05 |
| 11 | 75,571 | 36,017 | 0 | 3,159 | 114,747 | 25.20 | 12.01 | 0.00 | 1.05 | 38.26 |
| 12 | 97,362 | 33,792 | 0 | 9,91 | 132,145 | 32.55 | 11.30 | 0.00 | 0.33 | 44.18 |
| Total | 1,079,511 | 229,705 | 19,352 | 9,550 | 1,338,118 | 50.60 | 10.76 | 0.91 | 0.45 | 62.72 |

### A.2  VISUALIZATION OF TRACKING PERFORMANCE

As shown in Figs. A-2 to A-7, we compared six testing sequences W/ and W/O Event data.

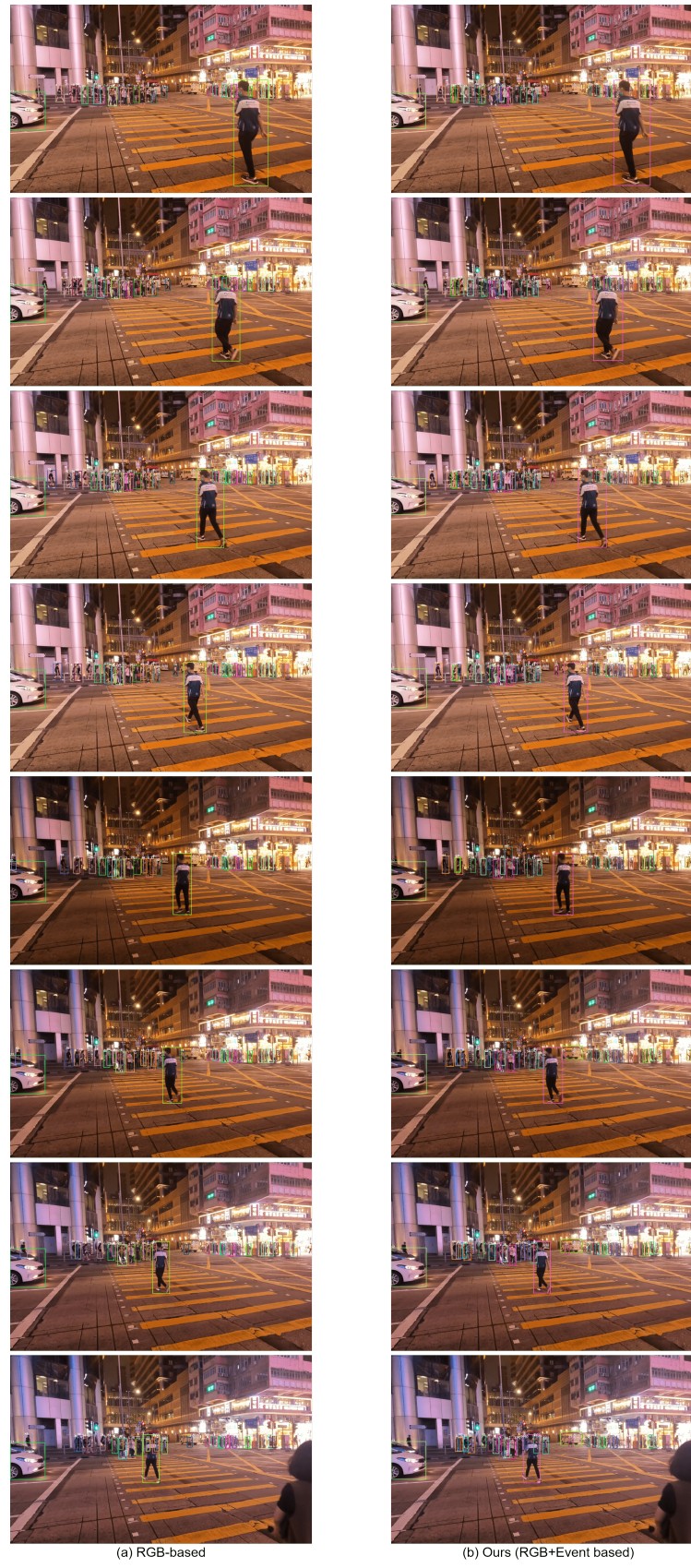

(a) RGB-based            (b) Ours (RGB+Event based)

Figure A-2: We present a qualitative comparison between a conventional RGB-based baseline method and our proposed RGB-Event approach on multiple frames within the *thrid* sequence.

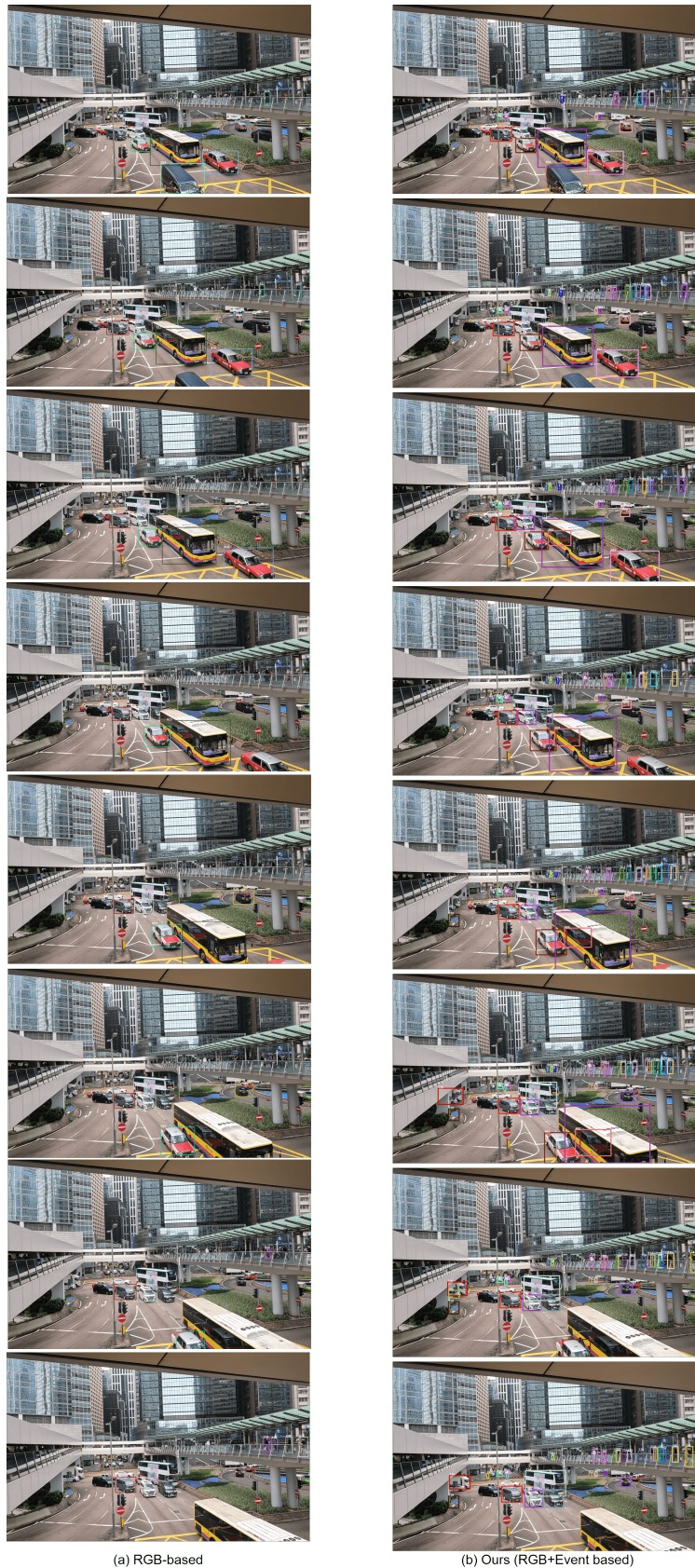

(a) RGB-based        (b) Ours (RGB+Event based)

Figure A-3: We present a qualitative comparison between a conventional RGB-based baseline method and our proposed RGB-Event approach on multiple frames within the *fifth* sequence.

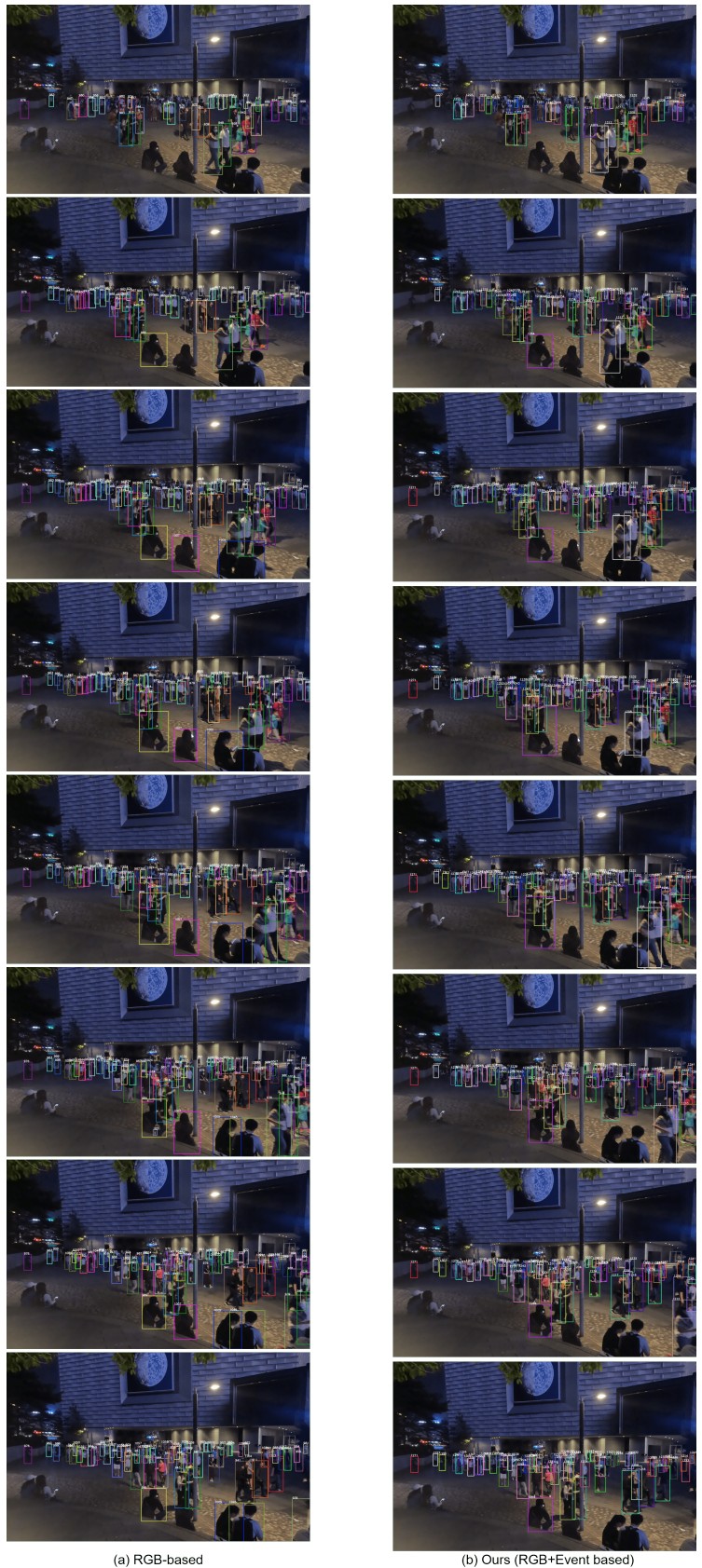

(a) RGB-based            (b) Ours (RGB+Event based)

Figure A-4: We present a qualitative comparison between a conventional RGB-based baseline method and our proposed RGB-Event approach on multiple frames within the *sixth* sequence.

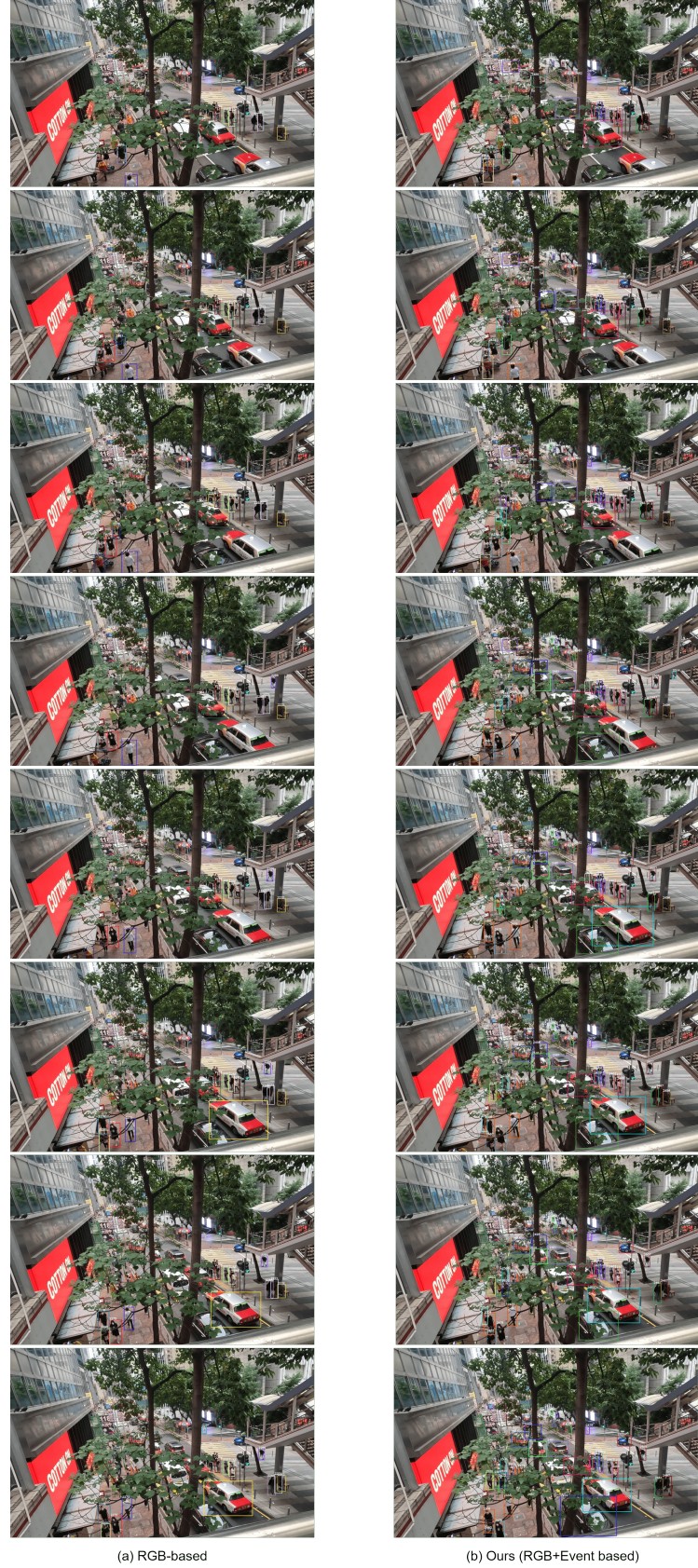

(a) RGB-based          (b) Ours (RGB+Event based)

Figure A-5: We present a qualitative comparison between a conventional RGB-based baseline method and our proposed RGB-Event approach on multiple frames within the *seventh* sequence.

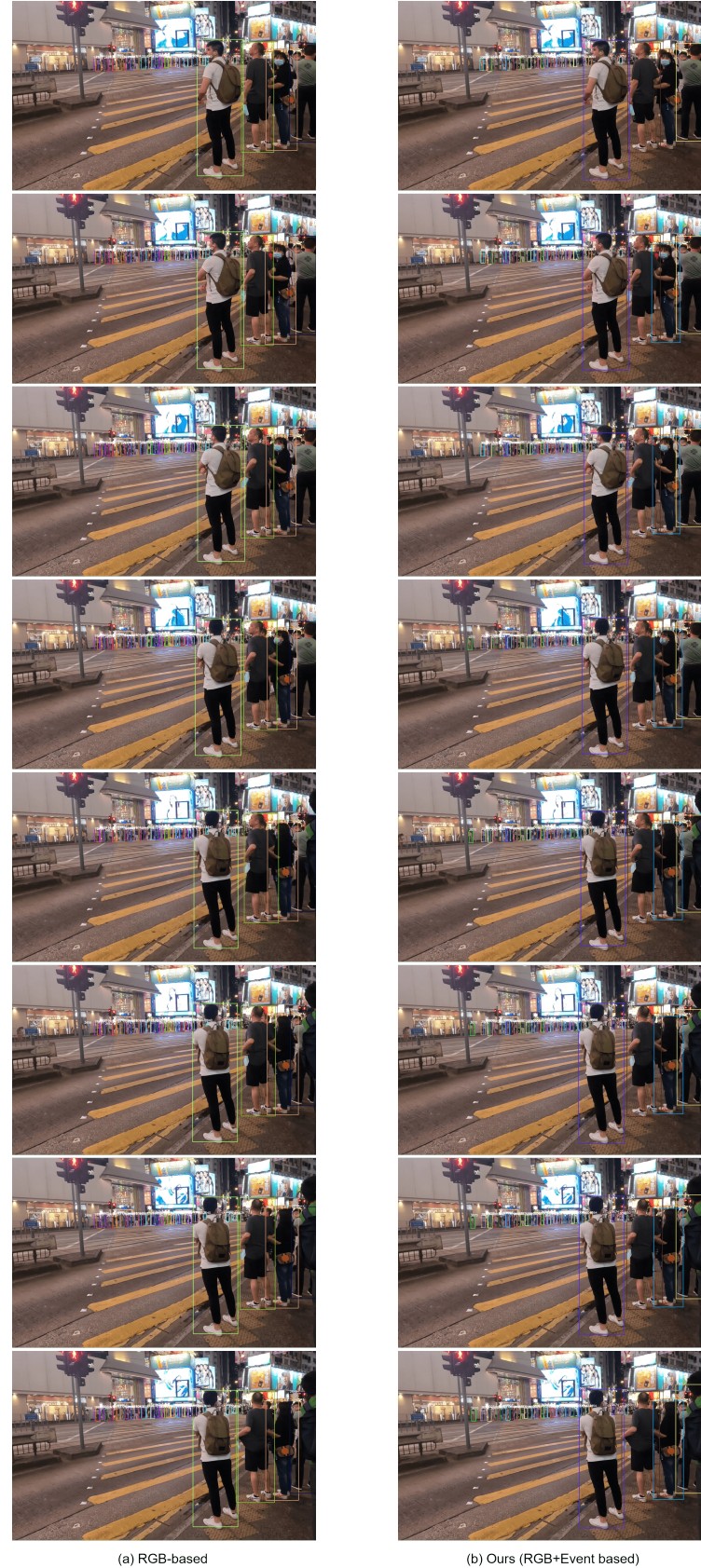

(a) RGB-based           (b) Ours (RGB+Event based)

Figure A-6: We present a qualitative comparison between a conventional RGB-based baseline method and our proposed RGB-Event approach on multiple frames within the *tenth* sequence.

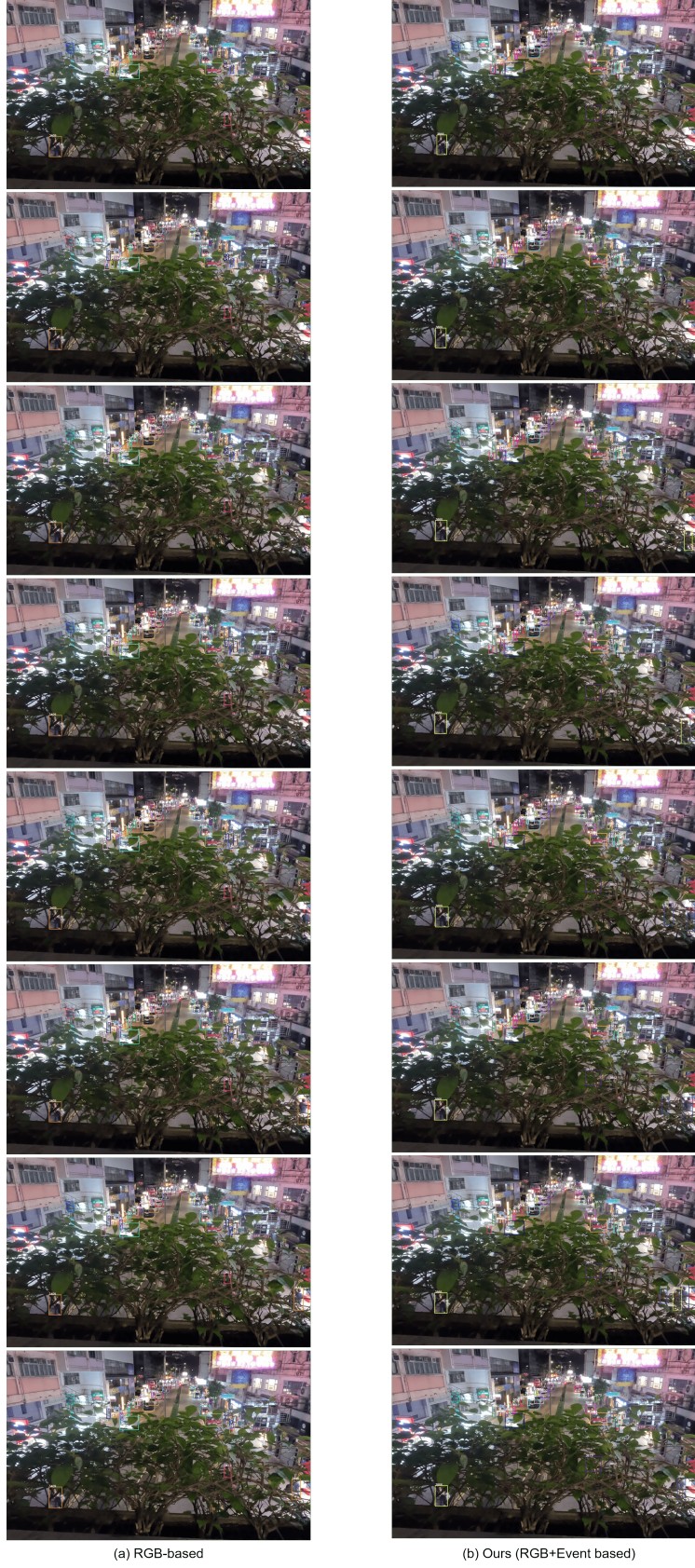

(a) RGB-based         (b) Ours (RGB+Event based)

Figure A-7: We present a qualitative comparison between a conventional RGB-based baseline method and our proposed RGB-Event approach on multiple frames within the *twelfth* sequence.

