# OpenReview forum: "RGB-Event MOT: A Cross-Modal Benchmark for Multi-Object Tracking"
_ICLR.cc/2024/Conference — ICLR 2024 Conference Withdrawn Submission_

### Official Review · Reviewer_QXZ6 · 2023-10-31

**Soundness:** 2 fair
**Presentation:** 3 good
**Contribution:** 2 fair
**Rating:** 5
**Confidence:** 5

**Summary:**

This paper introduces a novel cross-modal RGB-Event dataset for Multi-Object Tracking (MOT), aimed at addressing challenges in object tracking in complex real-world scenarios such as low-illumination conditions, small object detection, and occlusions. Utilizing the advantages of Event-based vision, known for its superior temporal resolution, vast dynamic range, and low latency, alongside conventional RGB data, the authors strive to advance the field of MOT. The newly developed dataset comprises nearly one million annotated ground-truth bounding boxes and is tested using state-of-the-art MOT algorithms, revealing a significant enhancement in performance with the integration of event data. The paper also explores the efficacy of different data fusion techniques, highlighting the potential of mask modeling over simple averaging. Through rigorous assessment and comparison with existing methods and datasets, the authors underline the potential of their proposed benchmark in driving further research and improving the robustness and versatility of detection and tracking systems, particularly in challenging visual scenarios. Besides, the authors acknowledge certain limitations of their dataset including static viewpoints and isolated hard cases, and suggest future directions for refining fusion techniques, embedding methods for event data, and development of specialized box association algorithms to better utilize the unique attributes of event data in MOT.

**Strengths:**

Here are some strengths of the paper:

	1. The paper is well-written and easy to understand. The authors provide clear explanations of the proposed algorithm and its components, as well as the motivation behind their approach.
	2. The paper introduces a unique cross-modal RGB-Event dataset for Multi-Object Tracking (MOT), significantly enriching the resources available for research in this field.
	3. The focus on overcoming practical challenges such as low-illumination conditions, occlusions, and small object detection aligns the paper with real-world needs in computer vision.
	4. Through thorough evaluation using state-of-the-art MOT algorithms, the paper substantiates the benefits of integrating event data with traditional RGB data.
	5. The authors intend to make the source code and the dataset publicly available upon acceptance, which fosters reproducibility and allows other researchers to build upon their work.

**Weaknesses:**

Here are some potential weaknesses:

	1. The exploration of data fusion techniques is somewhat limited with the utilization of simplistic averaging and mask modeling, which might not fully exploit the potential of cross-modal data fusion.
	2. The paper seems to focus on early fusion strategies, where RGB and Event data are fused at the input level. However, it does not explore or discuss middle or late fusion strategies, which could provide different perspectives and potentially better performance.
	3. The paper could have delved deeper into proficient embedding methods for event data, which is essential for leveraging the high temporal resolution of event data effectively.
	4. The paper does not delve into the discussion or evaluation of transformer-based methods for Multi-Object Tracking (MOT), which have been emerging as powerful tools for handling sequences and spatial relationships in data.
	5. The paper does not provide information or discussion on the frame rate (FPS) of the tracker after incorporating event data. This is crucial as the processing speed is a vital aspect of real-time multi-object tracking applications.
	6. The paper aims to optimize detection performance through the integration of RGB and Event data, yet lacks discussion or specification on the particular detector used. This omission can lead to a lack of clarity and could hinder the reproducibility of the proposed methods.

**Questions:**

1. Could the authors elaborate on why only simplistic averaging and mask modeling were chosen for data fusion over more sophisticated techniques?
	2. Why were middle or late fusion strategies not explored, and do the authors anticipate different outcomes with these alternative fusion strategies?
	3. Could the authors provide more details on the embedding methods explored for event data and their impact on the system's performance?
	4. Have the authors considered integrating transformer-based methods for multi-object tracking, given their promise in sequence processing tasks?
	5. Can the authors provide the frame rate of the tracker post event data integration, and discuss its implications for real-time application?
	6. Could the authors specify the detector used, its integration with RGB and Event data, and the influence of the choice of detector on the results?

---

> ### Author Response · Authors · 2023-11-18
> **Response to reviewer #QXZ6**
>
> Thank you for your feedback, reviewer #QXZ6. We'd like to address your concerns with the following clarifications:
> 1) Our contributions mainly lie in the construction of dataset and making current SOTA trackers work on the proposed dataset. Thus, authors expect that the reviewer could have a further consideration of the contributions on dataset construction.
> 2) Fusion in different stage indeed could affect the network performance. However, due to the large size of image, the early fusion-based method is actually the only possible methods. Fusion in the middle or late stage would consume much memory.
> 3) High temporal resolution is important for event data. Current experiment show that simply integrate event into the framework could indeed improve the performance. Moreover, the contribution is focus on dataset construction. We will explore more integration method in the future.
> 4) Actually, the first method, TrackFormer, in Table.4 is transformer-based method.
> 5) After integration with Event data the pipeline is just a bit slower about 25FPS compared to 30 FPS of original ByteTrack. For more efficient implementation, there are some libraries, e.g., ONNX.
> 6) We utilize the YOLOX as detector, which is quite popular choice for MOT algorithms.

---

> > ### Comment · Reviewer_QXZ6 · 2023-11-22
> >
> > Thank you for your detailed response, addressing some of the concerns I raised. However, I maintain my original rating. All the best!

---

### Official Review · Reviewer_FGic · 2023-10-31

**Soundness:** 3 good
**Presentation:** 3 good
**Contribution:** 1 poor
**Rating:** 3
**Confidence:** 5

**Summary:**

The paper first proposes the rgb-event multi-object tracking task which is new and interesting. It handles the low illumination, occlusion, and low-latency issues in the traditional rgb-based MOT task. It proposes a dataset that contains 12 videos for evaluation and also provides some baselines for future works to compare. For the baseline approach, the authors propose to fuse the dual modalities using concatenate or masking technique. This paper is well-written and the organization is good.

**Strengths:**

The paper first proposes the rgb-event multi-object tracking task which is new and interesting.

**Weaknesses:**

For the issues of this work:

the dataset is relatively small, 12 videos is not large-scale enough for current tracking tasks, especially in the big model era;
the baseline method is not novel, only simple fusion strategies are exploited; no novel fusion modules are proposed;
Therefore, I tend to reject this paper and encourage the authors to collect a larger rgb-event mot dataset or a more novel mot tracking framework.

**Questions:**

1. the dataset is relatively small, 12 videos is not large-scale enough for current tracking tasks, especially in the big model era;

2. the baseline method is not novel, only simple fusion strategies are exploited; no novel fusion modules are proposed;

---

> ### Author Response · Authors · 2023-11-18
> **Rresponse to reviewer #FGic**
>
> Thank you for your insightful feedback, reviewer #FGic. We'd like to address your concerns with the following clarifications:
> 1. Comparison with Existing Datasets: Our dataset is designed to be on par with established datasets such as MOT17 and MOT20. We recognize that MOT datasets often fall into two distinct categories: those with numerous sequences but fewer objects per sequence (like DanceTrack and SportsMOT), and those with fewer sequences but densely packed objects (such as MOT16 and MOT20). Our dataset aligns with the latter category. Given the practical challenges, it's unfeasible to annotate a large volume of videos with densely packed objects. However, as noted in our paper, we are committed to continually augmenting our dataset with additional sequences. We believe our dataset makes a significant contribution to the nascent field of event-based vision.
>
> 2. Focus of the Paper: The primary aim of our paper is to demonstrate the potential of event-based vision in enhancing the reliability of traditional RGB-based vision systems. We emphasize the value of our dataset as the main contribution to this field. Moreover, we note that it's time- and effort-consuming make such a cross-modal MOT framework work. We tried different training settings and hyperparameters. Thus, authors expect that the reviewer could have a further consideration of the contributions on dataset construction and baseline methods.

---

> > ### Comment · Reviewer_FGic · 2023-11-23
> >
> > Thanks for your response to my questions. However, I still think the dataset is not large and diverse enough for RGB-Event MOT, although it is the first dataset proposed for this task. More videos and new fusion strategies are still needed to be improved for this paper, I suggest.

---

### Official Review · Reviewer_2xGM · 2023-11-01

**Soundness:** 2 fair
**Presentation:** 3 good
**Contribution:** 2 fair
**Rating:** 3
**Confidence:** 5

**Summary:**

The paper proposes a dataset for combined RGB and Event camera tracking. The initial focus of the paper is to motivate the use of Event cameras for the task, considering challenges like low-illumination, occlusions etc. The dataset is then described in detail. The paper then applies existing algorithms post merging the RGB and Event camera data in the feature space. The results are presented on the proposed dataset.

**Strengths:**

- The dataset with combined and calibrated RGB and Event camera data is valuable. And the data collection + annotation is the major strength of the paper.

- The need for using Event camera is well motivated

- The paper is easy to read and understand

**Weaknesses:**

1. The method and experiments sections are vaguely presented. Several crucial details are missing:

(a) It is not clear, if a separate backbone (Figure 4) is used for both Event and RGB cameras. If yes, how were they trained?

(b) Was the proposed dataset used for training the detector?

(c) Was anything beyond the detector was trained or updated in the method? Was the retrained detector used in all the baseline methods?

(d) How was the Re-ID network trained? If not, which network was used for computing the Re-ID features?

(e) A common observation in several prior MOT paper is that Re_ID does not really play a significant role. The performance largely depends on the detection proposals and the motion model. An ablation without using the Re-ID features would be useful.

(f) The paper does not talk about the motion model at all. An ablation with and without using any motion model would add value to the paper.

(g) How exactly is averaging or masking done. Corresponding equations are warranted. It is extremely vague in the current form.



2. The description is unclear at several places

(a) What is e in Eqn1?

(b) If \delta is a scalar why does it vary with time (Eqn1 \delta_t)



3. If one uses consecutive frame differences instead of the frames from the event camera, will that achieve similar gains?

**Questions:**

Please address the concerns raised in the weaknesses section. The method section is completely unclear in the current form.

---

> ### Author Response · Authors · 2023-11-18
> **Response to reviewer #2xGM**
>
> Thank you for the feedback, reviewer # 2xGM. We'd like to address your concerns with the following clarifications:
> 1. a) All the event and RGB domains share a same ResNet backbone, we feed the RGB and Event data simultaneously into the first convolutional layer and then sum them up. We have supplemented the training code.
>  b) Yes, the proposed RGBEvent-MOT is utilized to train detector.
> c) Only the detector is trained by the proposed dataset. The retrained detector is utilized in Table 4 of none gray region methods, ByteTrack and BOT-SORT.
> d) The Re-ID network is not trained. For the BOT-SORT, we follow their instruct to utilize image feature from FastReID library.
> e) Actually, the ByteTrack utilizes the KalmanFilter to achieve box association. Through comparison the performance of BOT-SORT with ByteTrack, we could figure out that utilizing ReID feature may not help the MOT algorithm to achieve better performance.
> f) Authors agree with that modeling of motion cue is essential for MOT. However, due to that the event camera has very high temporal resolution, we need to redesign an event-aware motion information processing pipeline to adapt this pipeline. Actually, building this dataset and make current multi-object trackers work on this data is really time and effort-consuming. Thus, authors expect that the reviewer could have a further consideration of the contributions on dataset construction.
> 2.
> a) e in Eqn.1 represents a polarity of an event.
> b)Due to that the interval of two frame is actually not exactly fixed, the there is a small variation between different \delta.3.Acctually, the TrackFormer in table 4 considers the multi-frame information. However, it event has worse performance than others. It shows that in the hard cases presented by this paper, directly utilize image to extract temporal information is not enough.